# Comparison of Water Sampling between Environmental DNA Metabarcoding and Conventional Microscopic Identification: A Case Study in Gwangyang Bay, South Korea

**Dong-Kyun Kim** [1] , **Kiyun Park** [1], **Hyunbin Jo** [1] **and Ihn-Sil Kwak** [1,2,*]

1   Fisheries Science Institute, Chonnam National University, Yeosu 59626, Korea
2   Faculty of Marine Technology, Chonnam National University, Yeosu 59626, Korea
*   Correspondence: iskwak@chonnam.ac.kr; Tel.: +82-61-659-7148

**Abstract:** Our study focuses on methodological comparison of plankton community composition in relation to ecological monitoring and assessment with data sampling. Recently, along with the advancement of monitoring techniques, metabarcoding has been widely used in the context of environmental DNA (eDNA). We examine the applicability of eDNA metabarcoding for effective monitoring and assessment of community composition, compared with conventional observation using microscopic identification in a coastal ecosystem, Gwangynag Bay in South Korea. Our analysis is based primarily on two surveys at a total of 15 study sites in early and late summer (June and September) of the year 2018. The results of our study demonstrate the similarity and dissimilarity of biological communities in composition, richness and diversity between eDNA metabarcoding and conventional microscopic identification. It is found that, overall, eDNA metabarcoding appears to provide a wider variety of species composition, while conventional microscopic identification depicts more distinct plankton communities in sites. Finally, we suggest that eDNA metabarcoding is a practically useful method and can be potentially considered as a valuable alternative for biological monitoring and diversity assessments.

**Keywords:** coastal ecosystem; eDNA; metabarcoding; microscopy; monitoring and assessment

## 1. Introduction

Environmental DNA (eDNA) is defined as genetic material indirectly obtained from a wide variety of environmental samples (e.g., air, water, and soil), rather than directly sampled from macro- and micro-organisms [1]. Since a specific region of DNA sequences accommodates the information about the identification of specific organisms of interest, eDNA collected from an environmental sample encompasses a variety of species information in an ecosystem [2]. The idea of eDNA was initiated from extracting the nucleic acids of microbes directly from environmental samples [1,3–5].

Nowadays, DNA across diverse taxonomic groups has been widely searched in the context of genome projects [6,7]. The rapid advancement of molecular technology, such as amplification using polymerase chain reaction (PCR), facilitates applications of DNA-based approaches that highlight the capacity of analysis to detect a variety of macro- and micro-organisms within the same sample. DNA-based identification has been regarded as efficient alternatives in terms of both time and cost in ecological research [8,9]. This analytical technique can be applied either to a single species/taxon using specific primers or to multiple species/taxa using generic primers in accordance with research objectives. DNA metabarcoding is a rapid method for assessing biodiversity from environmental bulk samples. In particular, rapidly growing next-generation sequencing (NGS) techniques have recently allowed

comprehensive surveys for biological monitoring and assessment [8,10]. To this end, a growing body of literature has put special emphasis on the advantages of metabarcoding, highlighting its usefulness for ecological management [2,9,11–15]. Accordingly, a new type of DNA-based identification method has been developed as DNA metabarcoding, and widely introduced with plenty of applicable potentials for biological monitoring and assessment [16–18]. Specifically, eDNA metabarcoding has been newly proposed to assess the status (e.g., healthy, threatened, or degraded) of an ecosystem by detecting single (rare) and/or multiple (abundant) species in terms of biodiversity [12,13,19]. Despite the relatively short history, eDNA metabarcoding is appealing for monitoring and assessment of ecosystems due to its species detectability, cost and effort efficiency, and no environmental disturbance [18].

In coastal marine ecosystems, plankton communities play a pivotal role in food chain flow and biogeochemical cycles [20]. Particularly, zooplankton communities including both mero- and holo-zooplankton exert large influences on fish biomass and fisheries resources especially associated with juvenile growth [21]. Conventional microscopic identification (CMI) methods have mostly been used to estimate the richness and abundance of plankton communities in an aquatic ecosystem [22,23]. CMI might be limited in taxonomic identification, because the resultant data quality depends upon expertise and subjectivity of the scientists, and may cause disturbance to the habitat, and it is difficult to detect rare and endangered species [2,24]. In contrast, an eDNA analysis contains competitive advantages over CMI in detecting rare or invasive species [25]. In addition, given the high cost and large efforts for data collection and analysis in CMI, eDNA metabarcoding sheds light on efficient monitoring and assessment of a target ecosystem [8,18]. Furthermore, the rapid biological responses/changes to ambient physicochemical conditions lead to high demands on a new method that is fast and inexpressive, such as NGS-based metabarcoding [8]. Yet, the applications of eDNA have not been covered as widely as we wished, because of its short history, and to date have focused more on paleoecology and endangered species [13,19].

In the sense that the eDNA metabarcoding is highly appealing for finding cryptic aquatic species in biological monitoring and assessment, our study focuses on testing the potential of eDNA metabarcoding in order to monitor coastal plankton communities and assess biodiversity in comparison to CMI. Hence, the aim of our study is to identify spatial and temporal heterogeneity of plankton community dynamics in Gwangyang Bay of South Korea, characterizing predominant species and ambient water quality conditions. Finally, we discuss the potential values of eDNA metabarcoding as an alternative approach for ecological monitoring and rapid assessment.

## 2. Materials and Methods

### 2.1. Description of the Study Site

Gwangyang Bay is located in the south coast of Korean peninsula (Figure 1). In terms of morphological features of the bay, water depth varies from 10 m at the Seomjin River estuary to 50 m at the outer bay. The bay has a semi-diurnal tidal cycle. The bay receives a large discharge (ca. annually 2298 mega MT year$^{-1}$, equivalent to 72.8 m$^3$ s$^{-1}$) from Seomjin River [26]. It appears that a significant amount of nutrients (19.7 × 10$^3$ moles N day$^{-1}$, 0.1 × 10$^3$ moles P day$^{-1}$, 18.2 × 10$^3$ moles Si day$^{-1}$ in average) come to the bay from the Seomjin River catchment (ca. 5000 km$^2$) [27]. Since the Seomjin River estuary relative to the Korean river estuaries remains open without barrages, the water mass between river and ocean exchanges more actively. This dynamic condition of the bay tends to shape great primary productivity and high biological diversity. From both an ecological and economical points of view, Gwangyang Bay (ca. 450 km$^2$ from the estuary to the outer bay) is the most productive coastal area in Korea. Specifically, Jeonnam Province containing Gwangyang Bay comprised 71% (1,297,815 MT year$^{-1}$) of aqua-cultural resources in a national scale as of 2016 (KOSIS, [28]). In addition, a large industrial area (e.g., oil refineries and steel plants) near the bay can be regarded as a significant pollution source. Thus, the intermittent release of various pollutants might be another factor disturbing water quality and benthic sediments [29].

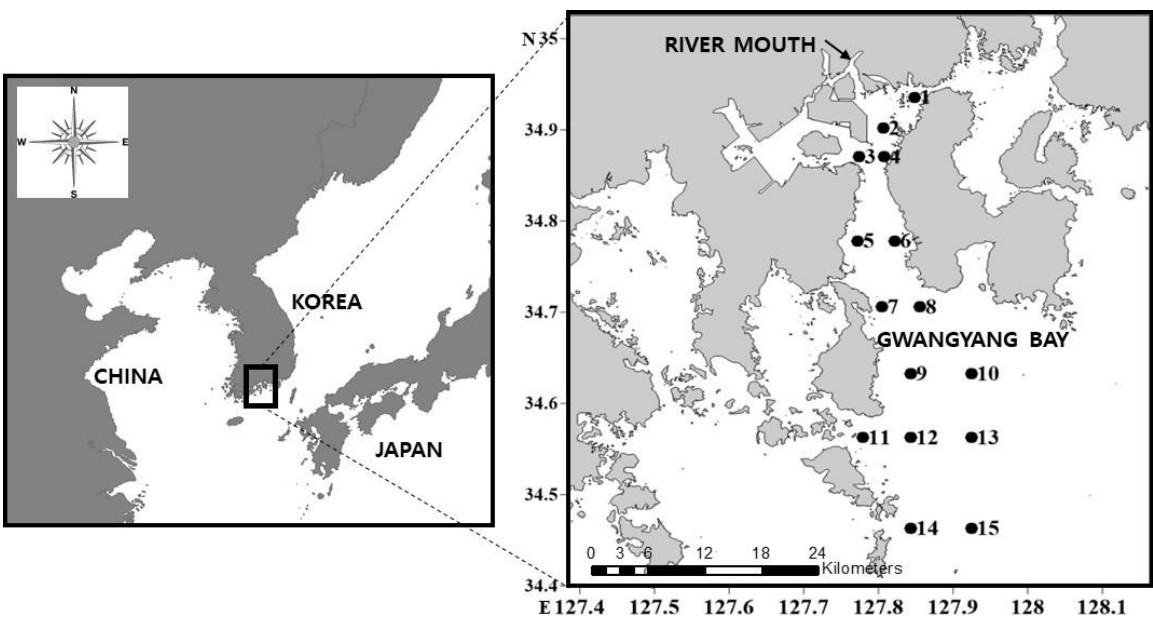

**Figure 1.** Map of the study sites (black closed circles) in Gwangyang Bay.

## 2.2. Sampling and Data Collection

The survey was conducted in June and September 2018, respectively. The total number of sampling sites was fifteen, and encompassed the extensive area from the Seomjin River estuary to the outer Gwangyang Bay (Figure 1). The water samples were collected vertically from sediments to surface (depth: 10–50 m). For marine plankton sampling, a 200 μm mesh-sized net was used. The corresponding water volume (ca. 7560 L; 7.56 $m^3$) was calculated by a flow-meter equipped in front of the net inlet. Zooplankton samples were identified and counted under a dissecting microscope (SV11, Zeiss and SZ60, Olympus, Tokyo, Japan), according to Chihara and Murano [30]. Water temperature and salinity were measured on site using a portable probe (Professional Plus, YSI, Yellow Springs, OH, USA). Nutrient and chlorophyll *a* concentrations (Chl-*a*) were analyzed in the lab using the collected water samples. Specifically for the measurement of phosphorus, nitrogen, and Chl-*a*, automatic water quality analyzer (AutoAnalyzer 3 HR, Seal Analytical Inc., Mequon, WI, USA) was used, and we adapted the standard analytical methods proposed by the Korea Ministry of Oceans and Fisheries (downloadable from http://www.mof.go.kr/jfile/readDownloadFile.do?fileId=MOF_ARTICLE_5689&fileSeq=1). For Chl-*a* measurement and eDNA metabarcoding, the water samples (1 L per sample) were immediately filtered in the lab, using a 0.45 μm pore-size membrane (MFS membrane filter, Advantec, Irvine, CA, USA). The membrane for Chl-*a* was then, homogenized after acetone extraction prior to the spectrophotometry. The membrane for eDNA was preserved at −80 °C. Organic and inorganic carbon concentrations were measured using a carbon analyzer (vario TOC cub, Elemetar, Langenselbold, Germany) on the basis of 850 °C combustion catalytic oxidation methods.

## 2.3. DNA Extraction and Metagenomic Sequencing

Genomic DNA was extracted by means of PowerSoil® DNA Isolation Kit (Cat. No. 12888, MO BIO, Germantown, MD, USA) in compliance with the manufacturers' protocol. Extracted DNA for sequencing was prepared according to the Illumina 18S Metagenomic Sequencing Library protocols (San Diego, CA, USA). DNA quantity, quality, and integrity were measured by PicoGreen (Thermo Fisher Scientific, Waltham, MA, USA) and VICTOR Nivo Multimode Microplate Readers (PerkinElmer, Akron, OH, USA). For our study, the 18S rDNA V9 barcode was used, because it has often been applied to semi-quantitatively estimate relative abundances within a sample [31–33]. More specifically, we obtained the primer information from a study by Guo et al. [33], which also followed the universal

primers for 18S V9 region designed by Amaral-Zettler et al. [34]. The primer sequences are as follows: 18S V9 primer including adaptor sequence (Forward Primer: 5′ TCGTCGGCAGCGTCAGATGTGT ATAAGAGACAG**CCCTGCCHTTTGTACACAC** 3′, Reverse Primer: 5′ GTCTCGTGGGCTCGGA GATGTGTATAAGAGACAG**CCTTCYGCAGGTTCACCTAC** 3′, the primers are in bold). The PCR master mixture of 25 μL (Macrogen Inc., Seoul, Korea) comprised 2 μL of genomic DNA (1 ng/μL), 1.25 μL of each primer (5 μM), 5 μL of 5 × Herculase II Reaction Buffer, 0.25 μL of dNTP mix (100 mM), 0.5 μL of Herculase II Fusion DNA polymerase (Agilent, Waldbronn, Germany), and 14.75 μL of PCR Grade water. To amplify the target region attached with adapters, as a first PCR process, the extracted DNA was amplified by 18S V9 primers with one cycle of 3 min at 95 °C, 25 cycles of 30 s at 95 °C, 30 s at 55 °C, 30 s at 72 °C, and a final step of 5 min at 72 °C for amplicon PCR product. As a second process, to produce indexing PCR, the first PCR product was subsequently amplified with one cycle of 3 min at 95 °C, 8 cycles of 30 s at 95 °C, 30 s at 55 °C, 30 s at 72 °C, and a final step of 5 min at 72 °C. A subsequent limited-cycle amplification step was performed to add multiplexing indices and Illumina sequencing adapters (Figure 2). The final products were normalized and pooled using the PicoGreen (ThermoFisher Scientific, Waltham, MA, USA), and the size of the libraries was verified using the LabChip GX HT DNA High Sensitivity Kit (PerkinElmer, Waltham, MA, USA).

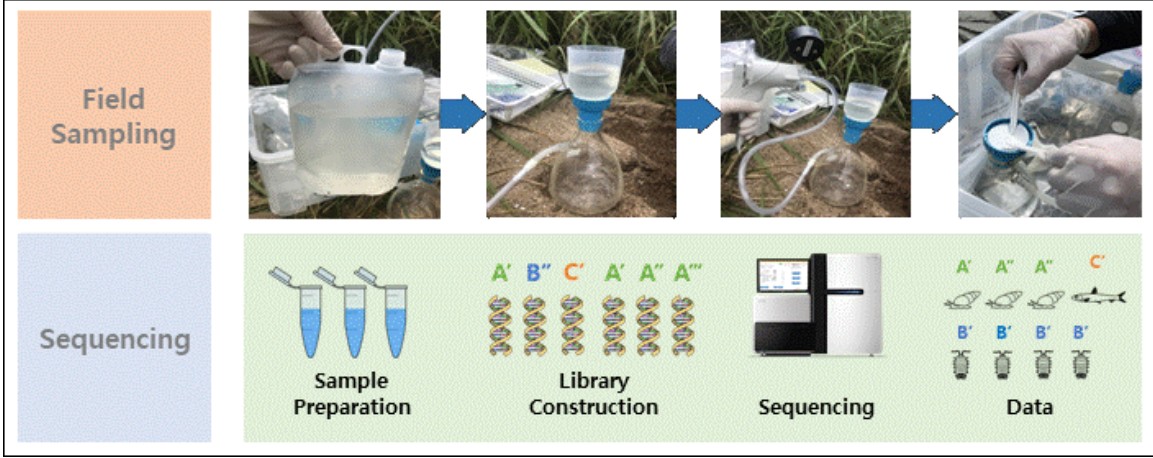

**Figure 2.** Analytical procedure of environmental DNA (eDNA) extraction and metagenomic sequencing.

A sequencing library is prepared by random fragmentation of the DNA or cDNA sample, followed by 5′ and 3′ adapter ligation. Alternatively, "tagmentation" combines the fragmentation and ligation reactions into a single step that greatly increases the efficiency of the library preparation process. Adapter-ligated fragments are then PCR amplified and gel purified. The PCR products were sequenced using the MiSeq™ platform (Illumina, San Diego, CA, USA) from commercial service (Macrogen Inc., Republic of Korea). In total, filtered 6,151,975 paired-end reads from the 30 samples were generated on the platform, of which 97.11% passed Q30 (Phred quality score > 30) in this study. Raw reads were trimmed with CD-HIT-OTU and chimeras were identified and removed using rDNATools. For paired-end merging, FLASH (Fast Length Adjustment of Short reads) version 1.2.11 was used. Each sample yielded paired-end reads ranging from 21,101–299,305 reads (mean: 180,940 reads), and all samples exhibited the saturation of the number of operational taxonomic units (OTUs) by rarefaction curve analysis (see Appendix A). Merged reads were processed and were clustered into OTUs using a bioinformatic algorithm, UCLUST [35], at a 97% OTU cutoff value (352 OTUs in gamma-diversity). The resulting 552 OTUs were classified into 19 genus-level taxonomic groups (those representing < 0.04% abundance were not plotted). Taxonomy was assigned to the obtained representative sequences with BLAST (Reference DB: NCBI—18S) [36] using UCLUST [35]. For the aforementioned processes of BLAST and UCLUST, we used an open-source bioinformatics pipeline for performing microbiome analysis, QIIME version 2 [37].

## 2.4. Analytical Methods

The self-organizing map (SOM) is an unsupervised neural network as machine learning, and it is commonly known as a powerful tool for pattern recognition from complex data [38]. In ecological research, the SOM has recently been considered as a more appropriate multivariate analysis than other conventional statistical approaches [39]. The SOM is robust and suitable in providing comprehensive views on highly complex and multi-dimensional data through reducing the data dimension. The efficiency of SOMs in information extraction was demonstrated across different hierarchical levels of life from molecules to ecosystems [40]. Several studies showed that the SOM was robust enough to capture the nonlinear pattern of an ecosystem [39,41,42]. For these reasons, the SOM has been extensively applied to pattern recognition in various ecological domains including benthic macroinvertebrates [43,44], plankton communities [45–48], dissolved organic matters [49], fish assemblages [50,51], and biomanipulation assessment [52,53].

In the SOM analysis, a total of 33 variables were used including six physicochemical parameters, 27 dominant plankton populations (10 from the eDNA, and 17 from the CMI samples). In selecting the number of variables, we only included the plankton communities, of which abundance was greater than 5% of the total abundance. That is, otherwise, the variables would contain too many zero values which could lead to topological biases in the SOM visualization. The SOM size was determined by the rule of $5\sqrt{\text{sample size}}$ [54]. The SOM model was developed using MATLAB 6.1 (The MathWorks Inc., Natick, MA, USA) and the SOM Toolbox (Helsinki University of Technology, Espoo, Finland).

For assessment of richness and diversity, the former simply equals to number of species, while the latter is based on the Shannon–Weaver index ($H' = -\sum p_i \ln p_i$, $p_i$ indicates a fraction of $i^{th}$ species) [55]. In calculating those biological indices, we excluded the taxonomical groups from eDNA samples, such as bacteria, mammals, reptiles, terrestrial plants, and amphibians, because the comparison between two different methods should be done at the same level of analytical resolution.

## 3. Results

### 3.1. Comparative Estimation of Coastal Biota between eDNA Metabarcoding and CMI

A variety of plankton communities were observed through the two identification methods in Gwangyang Bay. There were differences in the number of identified communities between the two methods (Table 1). In terms of quantity, eDNA metabarcoding seemed to be capable of detecting more species. The average numbers of observed (identified and unidentified) species from the eDNA samples were 27.9 (min to max: 20–36) in June and 49.8 (min to max: 13–72) in September, while those from the CMI were 19.6 (min to max: 12–23) and 18.9 (min to max: 12–24), in June and September, respectively (Table 1). Albeit comparing only with the identified species, we found that the number of species was higher in the eDNA samples than the CMI. On the other hand, in terms of the capability of identification of the eDNA metabarcoding, the unidentified species groups comprised 38% in June and 19% in September (Table 1). Accordingly, in Gwangyang Bay, the eDNA samples identified more species in a higher proportion in September.

In the eDNA samples, the richness values in September were as twice high as those in June (Table 1). The number of identified species was lower in June (mean ± S.D.: 20.2 ± 3.3) than in September (41.7 ± 15.6), which was quite consistent across the study sites. In addition, this pattern was similarly observed from diversity values of the eDNA samples (averages: 1.0 in June and 2.0 in September). The spatial variation of the richness was also lower in June (coefficient of variation: ca. 15%) than in September (ca. 40%). Namely, the heterogeneity of plankton distribution became large in late summer. On the contrary, in the CMI samples, the species richness did not differ between early and late summer; the values of mean and S.D. were 19.6 ± 3.1 in June, and 18.9 ± 3.5 in September. Notably, the level of diversity was comparatively higher in June (2.3 ± 0.2) than in September (1.6 ± 0.2), which was counter to the diversity pattern from the eDNA samples. Considered as a whole, the temporal changes

of biological communities seem to be more distinct, compared to their spatial variation. Nonetheless, we also note some discrepancy of the results in diversity between the two identification methods.

**Table 1.** Richness and Shannon diversity of the samples between water eDNA and conventional microscopic identification (CMI) in Gwangyang Bay. The numbers in the brackets indicate the number of unidentified groups.

| | June | | | | September | | | |
|---|---|---|---|---|---|---|---|---|
| | eDNA | | CMI | | eDNA | | CMI | |
| Site | Richness | Diversity | Richness | Diversity | Richness | Diversity | Richness | Diversity |
| GY1 | 28 (6) | 1.01 | 23 | 2.37 | 54 (9) | 1.99 | 20 | 1.62 |
| GY2 | 28 (7) | 1.65 | 23 | 2.43 | 59 (10) | 2.39 | 17 | 1.54 |
| GY3 | 20 (5) | 1.08 | 23 | 2.55 | 34 (5) | 1.50 | 12 | 1.34 |
| GY4 | 23 (6) | 1.35 | 22 | 2.46 | 72 (11) | 2.66 | 16 | 1.70 |
| GY5 | 25 (7) | 1.18 | 22 | 2.36 | 44 (7) | 1.65 | 13 | 1.35 |
| GY6 | 22 (6) | 0.72 | 20 | 2.33 | 62 (8) | 2.13 | 19 | 1.55 |
| GY7 | 29 (8) | 0.87 | 12 | 1.96 | 13 (3) | 0.24 | 18 | 1.78 |
| GY8 | 33 (10) | 1.08 | 18 | 2.24 | 29 (6) | 1.91 | 23 | 1.90 |
| GY9 | 35 (11) | 1.59 | 22 | 2.48 | 58 (8) | 2.31 | 24 | 1.61 |
| GY10 | 27 (8) | 1.05 | 20 | 2.37 | 48 (10) | 2.44 | 18 | 1.41 |
| GY11 | 36 (11) | 1.50 | 16 | 1.92 | 72 (11) | 2.69 | 18 | 1.72 |
| GY12 | 34 (10) | 0.56 | 18 | 2.31 | 35 (4) | 2.02 | 20 | 1.61 |
| GY13 | 31 (8) | 0.73 | 18 | 2.38 | 46 (8) | 1.57 | 23 | 1.81 |
| GY14 | 24 (6) | 0.64 | 18 | 1.75 | 45 (9) | 1.96 | 22 | 1.99 |
| GY15 | 24 (7) | 0.70 | 19 | 2.05 | 64 (12) | 2.20 | 20 | 1.66 |
| Mean | 27.9 (7.7) | 1.0 | 19.6 | 2.3 | 49.7 (8.1) | 2.0 | 18.9 | 1.6 |
| S.D. | 5.0 (1.9) | 0.4 | 3.1 | 0.2 | 17.8 (2.7) | 0.6 | 3.5 | 0.2 |

To evaluate the consistency of detection and identification of marine plankton groups, we compared the differences of community composition between eDNA and CMI samples (Figure 3a,b). Although various groups were detected by eDNA metabarcoding, the community composition was based on the identified groups in eDNA samples, in comparison with those from the CMI samples. In the higher rank of taxonomical classification (>phylum), the eDNA samples comprised 28% of phytoplankton (i.e., algae) and 15% of zooplankton (i.e., Copepoda) (Figure 3a), whereas the CMI samples showed 64% of zooplankton (Figure 3b).

In the eDNA samples, the dominant groups in phytoplankton were diatoms (e.g., *Thalassiosira* spp.) and dinoflagellates (*Hematodinium* spp.). In zooplankton, the dominant groups were marine calanoid copepods such as *Acartia* spp. and *Centropages* spp. in Gwangyang Bay. Crustacea occupied 17% of the identified species, and were primarily comprised of Amphipoda (e.g., *Caprella* spp.), Cirripedia (e.g., barnacles), and Decapoda (e.g., *Corophium* spp.). Cnidaria and Mollusca also engaged species richness of 24% in our study area (Figure 3a). The former consisted mainly of small polyp stony coral, such as *Acropora* spp., and the latter mostly comprised bivalves, such as *Crassostrea* spp. and *Musculista* spp. In addition, several groups, which were relatively low proportionally in CMI, were also well identified, including Annelida (5%), Chaetognatha (4%), Echinoderma (3%), and fish (4%). Particularly for fish, the identification of fish species was quite limited in the eDNA samples, and hence only three genera were identified (*Arnoglossus*, *Engraulis*, and *Oryzias* spp.).

By comparison, the CMI samples showed different proportion in species richness (Figure 3b). The main composition (64%) of zooplankton comprised Cladocera (e.g., *Evadne* spp. and *Podon leuckarti*) as well as Copepoda (e.g., 15 calanoid species and three cyclopoid species). Conversely, a limited number of phytoplankton was identified in the CMI samples, compared to the eDNA samples. The identified phytoplankton were mostly dinoflagellates which were mainly *Noctiluca scintillans*. Crustecea occupied 11% of species richness. Similar to the identified species from the eDNA samples, they were primarily composed of Amphipoda, Cirripedia, and Decapoda. However, most of them were in forms of larvae which was unable to be identified specifically in the CMI samples. Other specific groups were observed

in a small proportion (3%: Annelida, Chaetognatha, Cnidaria, and Echinoderma, and 5%: Fish and Mollusca, see Figure 3b). Nevertheless, in a finer resolution, there was some commonality of species groups between eDNA and CMI samples (Table 2). In both samples, several genera including *Acartia*, *Acropora* and *Centropages*, were commonly observed. At the Gwangyang Bay, *Acartia* spp. were commonly predominant in early summer, while *Centropages* spp. were relatively predominant in late summer. *Acropora* spp. were primarily observed from the eDNA samples around the inner bay in early summer. At the outer bay, including at site 14 and site 15, a dinoflagellate group of *Hematodinium* was relatively abundant, especially in the eDNA samples. In contrast, *Hematodinium* was not detected by CMI in the same area. Moreover, *Oithona* spp. were most predominant in this area, but were relatively less abundant in the outer bay, compared to the inner bay.

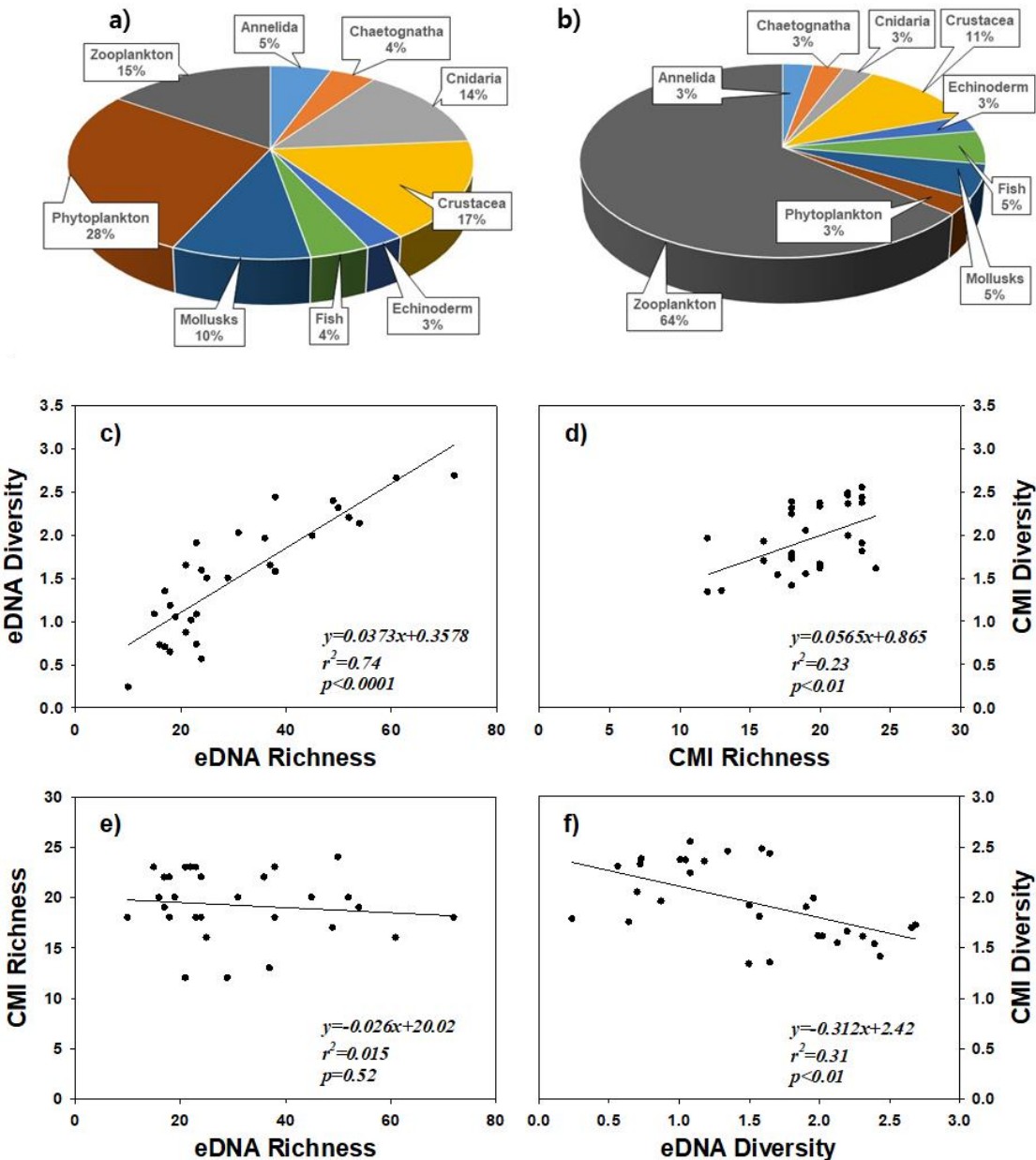

**Figure 3.** Community composition from the samples between (**a**) eDNA metabarcoding and (**b**) conventional microscopy identification (CMI). The scatter plots indicate the relationships between species richness and Shannon–Weaver diversity on (**c**) eDNA, (**d**) CMI, and those (**e**) between richness and (**f**) between diversity of the two methods, respectively.

**Table 2.** Dominant plankton groups observed during the summer season (June and September) in Gwangyang Bay.

| Site | eDNA Metabarcoding | CMI |
|---|---|---|
| GY1 | *Acropora, Candacia, Caprella, Oryzias* | *Acartia, Paracalanus* |
| GY2 | *Acropora, Candacia, Caprella, Corophium, Oryzias* | *Acartia, Corycaeus, Centropages, Corycaeus, Oithona, Paracalanus, Sagitta* |
| GY3 | *Acartia, Centropages* | *Acartia, Noctiluca, Oithona, Paracalanus, Sagitta* |
| GY4 | *Acartia, Acropora, Caprella, Corophium* | *Acartia, Corycaeus, Noctiluca, Oithona, Paracalanus, Sagitta* |
| GY5 | *Acartia, Acropora, Centropages* | *Acartia, Noctiluca, Paracalanus, Sagitta* |
| GY6 | *Acropora, Hematodinium* | *Acartia, Corycaeus, Noctiluca, Oithona, Paracalanus, Sagitta* |
| GY7 | *Acartia* | *Centropages* |
| GY8 | *Acropora, Caprella,* | *Centropages, Noctiluca* |
| GY9 | *Acartia, Acropora, Candacia, Centropages, Hematodinium* | *Centropages, Noctiluca* |
| GY10 | *Acropora, Thalassiosira* | *Centropages, Corycaeus, Sagitta* |
| GY11 | *Candacia, Caprella, Centropages* | *Centropages* |
| GY12 | *Centropages, Hematodinium,* | *Centropages, Paracalanus* |
| GY13 | *Candacia, Centropages* | *Centropages, Paracalanus* |
| GY14 | *Hematodinium* | *Oithona* |
| GY15 | *Hematodinium* | *Oithona* |

## 3.2. Relationships of Biotic Information between eDNA and CMI Samples

To examine consistency of biological information between different sampling strategies, the relationships between species richness and diversity were comparatively assessed. In both eDNA and CMI samples, species richness and diversity were positively correlated with each other (Figure 3c,d). The eDNA samples showed stronger signal of the positive relationship between species richness and diversity than the CMI samples, and the interpretability of species richness on corresponding diversity was three times higher in the eDNA samples ($r^2 = 0.74$) than in the CMI samples ($r^2 = 0.23$). Although both samples showed the significant relationships between the two, the relationship was clearer in the eDNA samples. On the other hand, we also examined the relationships between the richness values and between the diversity values (Figure 3e,f). There was no statistical significance between the richness values (i.e., eDNA versus CMI samples) (Figure 3e). In addition, although the diversity values exhibited statistical significance in their relationship, the signal was slightly negative, which was counterintuitive (Figure 3f). In consequence, it appeared that the information obtained from the same methodology was consistent enough to project the relationship between species richness and diversity. Conversely, it was found that there was a discrepancy of biotic information between eDNA and CMI samples.

## 3.3. Assessment of Biogeochemical Characteristics in Gwangyang Bay

The clustering analysis using the SOM characterized biogeochemical features of Gwangyang Bay into four distinct patterns. The four clusters determined by the SOM shaped spatiotemporal heterogeneity of the data samples at Gwangyang Bay (Figure 4 and Appendix B). It is remarkable to discern the spatiotemporal pattern that cluster 1 included site 1 to site 6 of June, cluster 2 site 7 to site 15 of June, cluster 3 site 1 to site 8 of September, and cluster 4 site 9 to site 15 of September as well as site 14 and site 15 of June (Figure 4a). In addition, the estimate of neighboring distances among the clusters indicated that the clusters were firstly separated as top (cluster 3 and cluster 4) and bottom (cluster 1 and cluster 2). As a consequence, the clustering result manifested that plankton community of Gwangyang Bay was primarily characterized by seasonal influences between early and late summer (i.e., June and September at Gwangyang Bay), and then was spatially distinguished. Strictly speaking, site 14 and site 15 of June were grouped as cluster 4 which represented the outer bay of late summer,

but they were placed on the bottom of cluster 4, which was characterized as the outer bay of early summer. Namely, these two sites appear to represent similar features on coastal plankton community, regardless of temporal changes in summer.

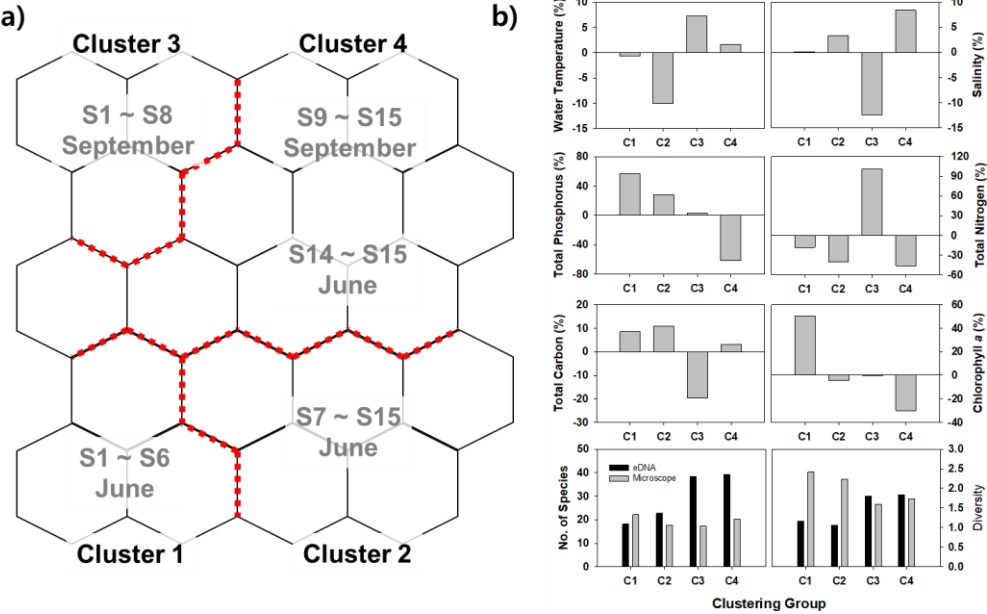

**Figure 4.** Clustering result (**a**) of the data of water eDNA and CMI based on the self-organizing map. The right panels (**b**) present the corresponding physical, chemical, and biological conditions in Gwangyang Bay. The horizontal lines of zero indicate corresponding grand average values (water temperature: 25.6 °C, salinity: 29.3 psu, TP: 0.049 mg L$^{-1}$, TN: 0.45 mg L$^{-1}$, TC: 22.4 mg L$^{-1}$, Chl-*a*: 4.36 mg L$^{-1}$).

Several water quality parameters delineated ambient physicochemical conditions associated with plankton community in Gwangyang Bay (Figure 4b). Water temperature was relatively lower in cluster 2 and higher in cluster 3 among the four groups. The higher salinity of the outer bay matched well with cluster 2 and cluster 4. Cluster 3 represented the inner bay of the summer, exhibiting lower salinity was higher water temperature. Concerning nutrient concentration, total phosphorus (TP) concentrations were higher in June (cluster 1 and cluster 2 in Figure 4) than in September (cluster 3 and cluster 4 in Figure 4). In the spatial scale, TP was higher at the inner bay (cluster 1 and cluster 3 in Figure 3) than at the outer bay (cluster 2 and cluster 4 in Figure 4). In addition to TP, total nitrogen (TN) concentrations were conspicuously high in cluster 3, which represented the inner bay in late summer. Total carbon (TC) concentrations displayed opposite patterns against TN. Among the four clustering groups, chlorophyll *a* (Chl-*a*) concentrations were highest at the inner bay in early summer, while were lowest at the outer bay in late summer. In view of biotic information, the number of species was relatively higher in cluster 3 and cluster 4 (September) based on the eDNA samples, while the diversity indices were comparatively higher in cluster 1 and cluster 2 (June) based on the CMI samples (Figure 4).

## 4. Discussion

### 4.1. Congruence of Taxonomic Information between eDNA Metabarcoding and CMI

Many of recent studies have strived to profile and quantify taxonomic composition of plankton communities using either eDNA metabarcoding or CMI [14,24,56]. Among them, a few studies have reported a degree of disagreement between the two pronged identification methods [32,57]. In this respect, our study also presented some disagreement, between eDNA and CMI samples, in community

composition (Figure 3a,b as well as in relationships of biotic information (Figure 3e,f). Some pieces of literature on eDNA monitoring have enumerated possible reasons to explain the discrepancy between the two identification methods. It is reported that the capacity of identification between molecular and morphological datasets could have mainly caused the disagreement [24,58]. That is, specimen identification can vary along accuracy of molecular reference databases [59]. Therefore, the establishment of well-curated databases of reference DNA sequences for identified specimens is essential in the field of eDNA metabarcoding to make the taxonomic information congruent with CMI. Additionally, there is another concern with the drawback of eDNA metabarcoding associated with technical biases/difficulties, such as copy number variation in the process of polymerase chain reaction (PCR) [60]. Related to a primer, its amplification and binding affinity are critical factors to bring about taxonomic biases in eDNA detection [61–65]. In terms of sensitivity of species detection, CMI-based assessment is also subject to an unpredictable, but probably significant, bias due to the presence of cryptic species [66]. Particularly in our study, marine calanoid copepods, *Candacia*, were only detected by eDNA metabarcoding in a very low proportion of <5%. However, we also admit that taxonomic misclassification due to lack of expertise and difficult to impossible taxonomic determination rather than just cryptic species also causes bias.

With these concerns in mind, our results on the community composition might be influenced by the primer amplification effects (Figure 3a,b). The previous related research reported some technical biases against low-abundant taxa in delineating microbial diversity [63]. In fact, while Cnidaria comprised 3% in CMI, they were 14% in eDNA samples. Likewise, Mollusks occupied 5% in CMI, but did 10% in eDNA samples (Figure 3a,b). In contrast to these differences, the compositional changes between the two samples were not significant for the rest low-abundant taxa containing Annelida, Chaetognatha, Echinoderm, and fish (Figure 3a,b). Namely, our results showed that low-abundant taxa could always be overestimated in eDNA metabarcoding. These results of difference and variation might be associated with several reasons. Firstly, eDNA metabarcoding is highly sensitive to detecting species. This high sensitivity is advantageous in identifying low-abundant/cryptic species. However, it can also lead to variations originating not only from organisms that are a few miles away from the sampling site but also from food items hidden in organisms. In addition, abundance estimates are possibly erroneous because many small organisms could generate the same number of sequence reads as a few large organisms. Secondly, although it is relatively unexplored, the copy number variation derived from the technical bias during the PCR process is another factor leading to inaccurate estimation [60,63]. Lastly, CMI is also error-prone depending on expertise/experience and specimen size. Therefore, we notice that eDNA may not be able to fully present diversity yet.

Despite some discrepancy between the eDNA and CMI samples, one highlighting point is the relational consistency in richness and diversity. Traditionally, plankton community assessment on richness and diversity has been complicated and time-consuming. However, compared to CMI, the eDNA metabarcoding also presented a positive relationship between richness and diversity (Figure 3c,d). Furthermore, while CMI exhibited a shorter range of richness and diversity (Figure 3d), the eDNA metabarcoding displayed a wider range (Figure 3c). Although its accuracy is another issue as previously mentioned, therefore, our study explicitly accounts for better capability of detection and identification by means of metabarcoding skills.

### 4.2. Potential Values of an eDNA Approach for Biological Monitoring and Assessment

Most conventional approaches for biological monitoring and assessment were based primarily on microscopy. Due to the time consumption and expertise requirement for identification in species level, the current environmental monitoring and assessment of community composition highly demand new alternative technologies in terms of cost efficiency. In this regard, eDNA metabarcoding has been deemed as a promising tool for species detection and identification [58]. Particularly in plankton research, the eDNA approach helped reveal a previously hidden taxonomic richness for diverse meroplankton, such as Bivalvia, Gastropoda, and Polychaeta, which are relatively hard to identify

in CMI [67]. Our study also advocates that a wider variety of species, including the aforementioned meroplankton, were detected in the eDNA samples (Table 1).

At the same time, however, we recognize that some discrepancies of abundance between metabarcoding and CMI have been contentious [62,68,69]. This discrepancy may limit the scope of eDNA research, which is also associated with the varying lengths of time to eDNA degradation in response to ambient environmental conditions [11,59,70,71]. Nevertheless, several studies have found a significant relationship between determining relative or rank abundance, highlighting the potential value of eDNA, though the variation inherent in environmental samples makes it difficult to quantify [12,32].

In our study, we found some clear patterns of coastal plankton communities in time (early vs. late summer) and space (inner vs. outer bay). From our analysis using eDNA and CMI samples, the main features of Gwangyang Bay could be characterized more clearly: (*i*) inner bay in early summer; (*ii*) outer bay in early summer; (*iii*) inner bay in late summer; and (*iv*) outer bay in late summer (Figure 5). Each characteristic was explicitly delineated by the prominent species. For example, in Gwangyang Bay, Asterozoa were predominant in early summer, *Sagitta* spp. were abundant in the inner bay, and zooplankton *Centrophages* spp. were in late summer. Dinoflagellates were separately characterized by *Noctilluca* spp. in early summer and by *Hematodinium* spp. in late summer. Although we did not use the eDNA samples solely, our spatiotemporal analysis presented the main plankton community features based on both eDNA and CMI samples. The CMI samples in addition to the eDNA make our pattern analysis more robust and reliable, because the predominant plankton would be separately presented if the eDNA and CMI samples differed significantly from each other. Thus, the information from the eDNA and CMI samples was highly similar given the subtle discrepancy of richness, diversity and their relationships (Figure 3c,d). However, we stress that the eDNA samples were good enough to delineate spatial and temporal characteristics of coastal plankton communities in Gwangyang Bay (Figure 5).

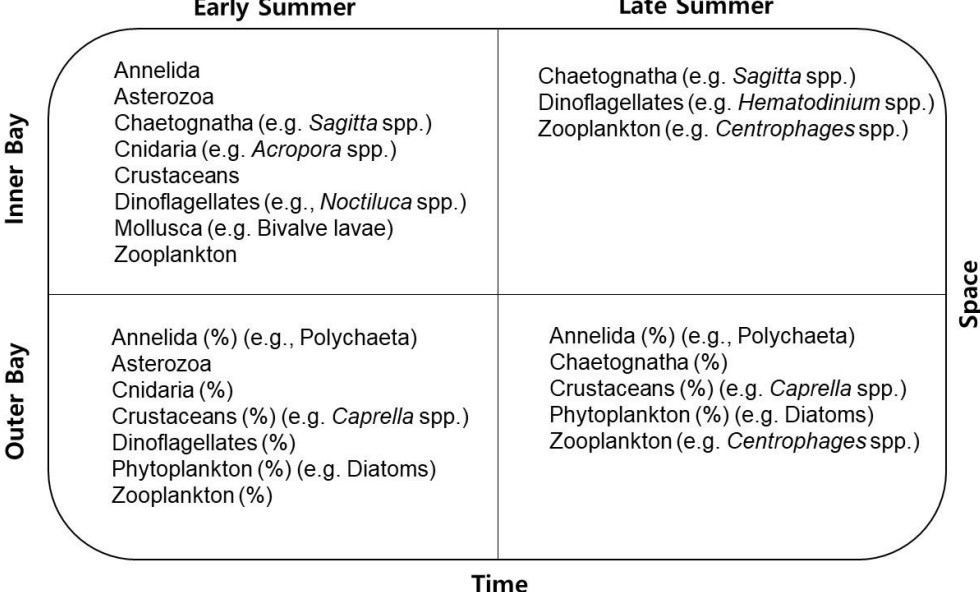

**Figure 5.** Main characteristics of marine plankton communities in Gwangyang Bay. The groups with % present relative abundance (derived from eDNA), and those without % present absolute abundance (derived from CMI).

In sum, we learn from our study that eDNA metabarcoding can be an effective alternative to monitor and assess entire communities from even a single sample. In addition, the eDNA metabarcoding is highly beneficial in terms of sensitivity for cryptic species and cost-efficiency for morphological

identification. At the same time, however, our study also put emphasis on bio-assessment that can be affected by some information discrepancy of richness and diversity between eDNA and CMI samples. Hence, eDNA-based research should be further investigated to make the derived results become more stable. The current limited capacity of eDNA-based research is probably subject to a great deal of uncertainties associated with amplification, reference database, NGS-sequencing, and eDNA degradation [57,71]. To this end, we stress that eDNA research should be more active in order to shed light on ecosystem monitoring and assessment in future.

**Author Contributions:** Conceptualization: D.-K.K., K.P., H.J. and I.-S.K.; Methodology: D.-K.K. and K.P.; Formal Analysis: D.-K.K. and H.J.; Investigation: D.-K.K., K.P. and H.J.; Resources: I.-S.K.; Writing—Original Draft Preparation: D.-K.K.; Writing—Review and Editing: D.-K.K. and I.-S.K.; Supervision: I.-S.K.; Project Administration: I.-S.K.; Funding Acquisition: I.-S.K.

**Funding:** This research was funded by the National Research Foundation of Korea, grant number NRF-2018R1A6A1A03024314.

**Conflicts of Interest:** The authors declare that they have no competing interests.

**Appendix A  Rarefaction Curves of the 18S rDNA V9 Samples in May (A) and September (B)**

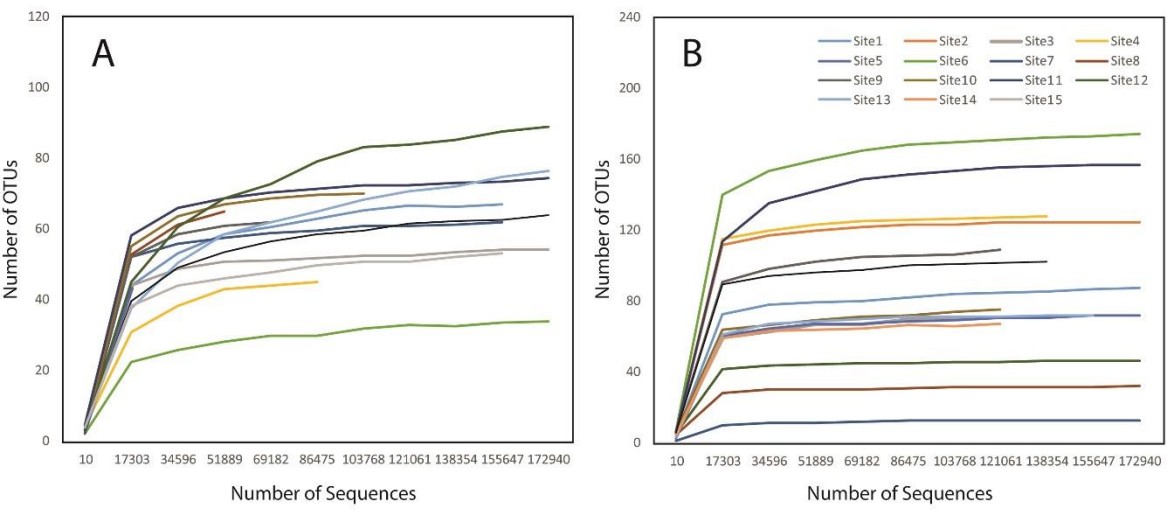

## Appendix B  Visualization of Explanatory Variables Derived from Self-Organizing Maps

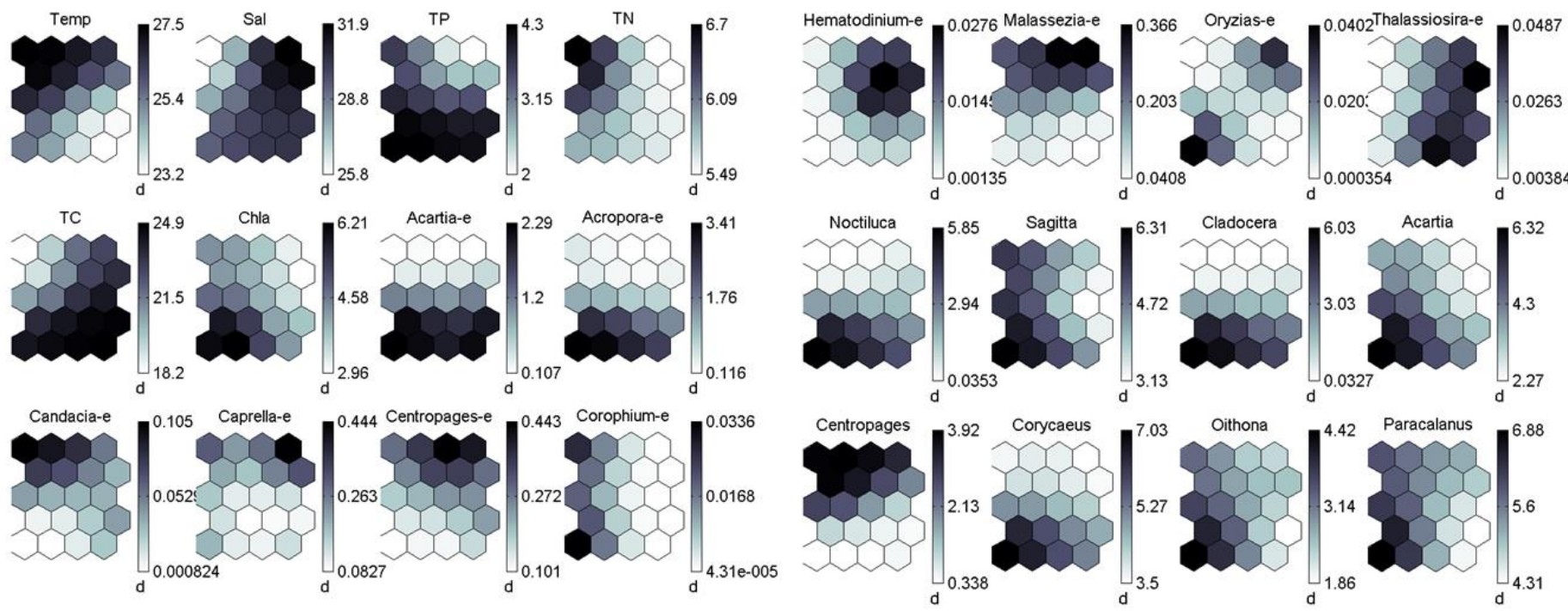

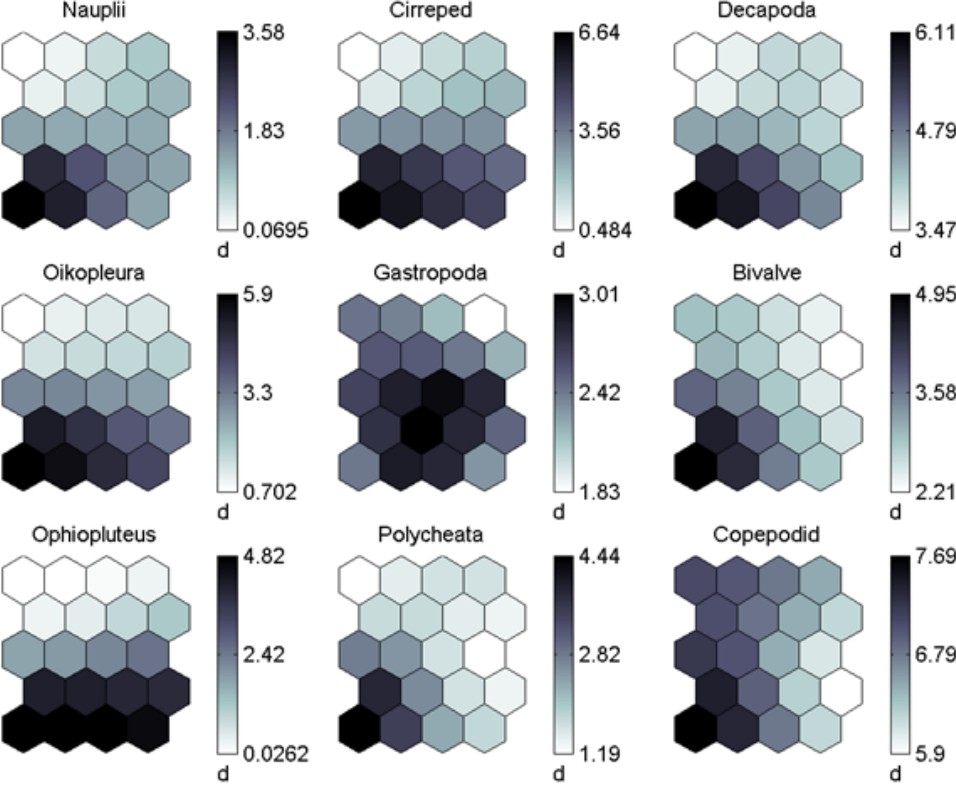

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
