# Peer review of "Comparison of Water Sampling between Environmental DNA Metabarcoding and Conventional Microscopic Identification: A Case Study in Gwangyang Bay, South Korea"

_applsci, doi:10.3390/app9163272_

Round 1

Reviewer 1 Report

In this study the authors investigated applicability of the eDNA metabarcoding for effective monitoring and assessment of community composition, comparing with conventional observation using microscopic identification in a coastal ecosystem in South Korea. They found that eDNA metabarcoding  gave a wider variety of species composition, while conventional microscopic identification depicts more distinct plankton communities in site.

This work is interesting  and novel. However, I suggest the following to be added:

1-some light or electronic microscopic photos showing plankton communities.

2-A diagram showing all steps of getting eDNA and its amplification.

3- Is there any similar studies in South Korea? If so, please compare with your study.

Reviewer 2 Report

Dear Dong-Kyun Kim and co-authors,

Overall, the manuscript is well written and structured. Kim and co-authors present very interesting new data and the objectives have been met in the results and discussion. The only critical aspect I noticed is that diversity/richness was compared between different samples and different methods without rarefaction analysis (or maybe without mentioning it), which is a standard method in ecology. In my experience, the number of sequence counts of different PCR products can differ in 2 and more orders of magnitude between each other and usually sequencing counts are much larger than conventional microscopy counts. Hence, diversity can only truly be compared on a similar basis of counts. With larger sample sizes, especially this different, more diversity will be captured. I strongly recommend including this analysis (e.g. R vegan package, I think it is also implemented in qiime). I have a few minor recommendations that I hope can improve the manuscript, below.

Kind regards!

Material and methods

Line 92: Please provide a citation for the standard protocol.

Line 118: I was wondering why you sequenced cloned PCR products without mentioning cloning beforehand. I believe this is a mistake? Otherwise, please give the details about the procedure.

Did you analyse PCR replicates or did you pool replicates? Did you include negative controls?

Please, describe how you calculated the Shannon-Diversity and the statistical methods you used for figures 2c-f. Also explain a little bit more clearly, what the difference between richness and Shannon Diversity is and why you used these diversity indices.

Results

While I can generally comment on the methods, results and discussion, I am not familiar with self-organizing maps. The Appendix Figure really helped me understanding the meaning of Figure 3 and contains very interesting information. I would recommend you specifically refer to your Appendix in the text. However, I am still not really clear, if the figure is the result of the DNA signal, of the CMI signal or of both combined. Maybe one more sentence about this could clarify this.

Fig. 2 e+f: Please indicate on x-and y-axis the method

Lines 226-228: To me, the message is not really clear. Maybe you could re-write the sentence: “Consequently, we summarized that the relationship between species richness and diversity was positively consistent within the samples obtained from the same methodology.”

Line 233: “It is remarkable…” does that refer to the spatio-temporal pattern itself or only to the first cluster? Maybe re-write it a little bit?

Fig. 3b: I don’t really understand the content of the graphs, because it shows relative changes. What exactly does a 5% change in water temperature (for example) represent? It could be 0.5°C or maybe 5%.

Discussion

I miss an interpretation of the diversity results with regard to the Shannon diversity and richness.

Line 280: I would not go as far as talking of bias. It is just a limit of the resolution capacity. Do the metabarcoding results indicate the presence of cryptic taxa? If yes, you could include this here.

Line 297-298: Do not repeat the results here (the r2 values in brackets can be removed).

Line 336-344: This paragraph will be clearer when shortened. Also be more specific. You can always say it should be further investigated, it would be more interesting if you mentioned in which direction, or how you will improve the technique.

In my opinion Figure 4 comes a little late and should be moved between the sections 4.1 and 4.2.

Minor corrections:

Line 30: “about identify” change to “about the identification”

Line 120: “were identified using chimeras” change to “chimeras were identified and removed”

Line 123: remove “into OTUs” and change “at a 97% OUT” to “at a 97% OTU”

Line 180: change “Mullusca” to “Mollusca”

Reviewer 3 Report

See attached file.

Reviewer 4 Report

The idea is good and I really appreciated the attempt to use SOMs in this kind of studies, but there are serious errors in the formulation of the article that demeans the real scientific potential.

The materials and methods are written inaccurately in many points, first of all, the bioinformatic and statistical analysis part. No one parameter is specified.

Missing sequencing results (OTU number) and the number of assignments obtained (confidence?). Without these two results, the data reported are totally meaningless.

Moreover, the authors discuss the spatial and temporal distribution of collected taxa but there is not a clear figure that shows these data (also the other figures have not clear labels)

These are specific corrections for the introduction, the other section needs a profound and substantial revision of the text. I will be happy to review the manuscript once it has been rewritten in these parts.

line 34 replace "a great deal of" with "plenty of"

line 55-69 Please rephrase the sentences more correctly (From a grammatical point of view).

line 61 In this sentence there is no real information, it is an opinion? If yes reformulate, if not provide more data and references.

line 66-67 15 sampling points are not enough to generalize data, please provide the exact name of the coastal bay. 

line 68-69 As in the previous comment, data are not enough to generalize. The authors have to specify in which case.

line 75 So yes or not? Please provide a reference for the data

###

Please provide in some section the data accessions in a public repository

Round 2

Author Response

All authors are grateful to anonymous reviewer(s) for providing valuable comments on our manuscript. We have done our best to improve the manuscript by respectfully accepting every single point raised by the reviewer(s). Please see attached.
